# The Association between Immune Checkpoint Proteins and Therapy Outcomes in Acute Myeloid Leukaemia Patients

**DOI:** 10.3390/cancers15184487

**Published:** 2023-09-09

**Authors:** Lukasz Bolkun, Marlena Tynecka, Alicja Walewska, Malgorzata Bernatowicz, Jaroslaw Piszcz, Edyta Cichocka, Tomasz Wandtke, Magdalena Czemerska, Agnieszka Wierzbowska, Marcin Moniuszko, Kamil Grubczak, Andrzej Eljaszewicz

**Affiliations:** 1Department of Haematology, Medical University of Bialystok, 15-276 Bialystok, Polandjaroslaw.piszcz@umb.edu.pl (J.P.); 2Department of Regenerative Medicine and Immune Regulation, Medical University of Bialystok, 15-269 Bialystok, Polandalicja.walewska@umb.edu.pl (A.W.); marcin.moniuszko@umb.edu.pl (M.M.);; 3Department of Haematology, Rydygiera Hospital in Torun, 87-100 Torun, Poland; edyta.cichocka@wszz.torun.pl; 4Department of Lung Diseases, Neoplasms and Tuberculosis, Nicolaus Copernicus University in Torun, 85-326 Bydgoszcz, Poland; tomasz_wandtke@wp.pl; 5Department of Hematology, Medical University of Lodz, 93-510 Lodz, Polandagnieszka.wierzbowska@umed.lodz.pl (A.W.); 6Department of Allergology and Internal Medicine, Medical University of Bialystok, 15-276 Bialystok, Poland; 7Tissue and Cell Bank, Medical University of Bialystok, 15-269 Bialystok, Poland

**Keywords:** acute myeloid leukaemia, CTLA-4, PD-1, PD-L1, B7-H3, cytarabine, daunorubicin, cladribine

## Abstract

**Simple Summary:**

Despite promising results of clinical trials, the use of immune checkpoint inhibitors (ICI) in acute myeloid leukaemia (AML) remains limited. To date, the United States Food and Drug Administration (FDA) has approved two PD-1 inhibitors, namely nivolumab and pembrolizumab, for treating relapsed/refractory classical Hodgkin lymphoma, and pembrolizumab in primary mediastinal large B-cell lymphoma. In AML, the potential of ICM inhibitors, namely PD-1, PD-L1, and CTLA-4 blockers, was confirmed in clinical trials in relapsed or refractory disease or in high-risk patients. However, the response rate varied widely, possibly due to the heterogeneity of the ICM expression level within different AML cases. Flow cytometric analysis was used for analysing PD-1, PD-L1, CTLA-4, and B7-H3 in untreated AML patients stratified on the basis of clinical outcome and cytogenetic molecular risk. Here, we demonstrated association of selected ICI in AML patients with their response to therapy and overall survival.

**Abstract:**

The development of novel drugs with different mechanisms of action has dramatically changed the treatment landscape of AML patients in recent years. Considering a significant dysregulation of the immune system, inhibitors of immune checkpoint (ICI) proteins provide a substantial therapeutic option for those subjects. However, use of ICI in haematological malignancies remains very limited, in contrast to their wide use in solid tumours. Here, we analysed expression patterns of the most promising selected checkpoint-based therapeutic targets in AML patients. Peripheral blood of 72 untreated AML patients was used for flow cytometric analysis. Expression of PD-1, PD-L1, CTLA-4, and B7-H3 was assessed within CD4+ (Th) lymphocytes and CD33+ blast cells. Patients were stratified based on therapy outcome and cytogenetic molecular risk. AML non-responders (NR) showed a higher frequency of PD-1 in Th cells compared to those with complete remission (CR). Reduced blast cell level of CTLA-4 was another factor differentiating CR from NR subjects. Elevated levels of PD-1 were associated with a trend for poorer patients’ survival. Additionally, prognosis for AML patients was worse in case of a higher frequency of B7-H3 in Th lymphocytes. In summary, we showed the significance of selected ICI as outcome predictors in AML management. Further, multicentre studies are required for validation of those data.

## 1. Introduction

The introduction of immune checkpoint inhibitors to cancer therapy was one of the most significant steps in previous years, which significantly improved the therapeutic landscape for patients with solid malignancies such as non-small-cell lung cancer, urothelial carcinoma, renal cell carcinoma, head and neck squamous cell carcinoma, and melanoma [1]. That field constitutes one of the most promising directions also in the context of AML management, apart from CAR T cells, bispecific T cell engaging antibodies (BiTEs) or dendritic cell vaccines. The need for novel drugs is predominantly associated with a still high number of patients with poor prognosis to standard therapy [2,3].

Considering the prevalence of the immune checkpoint inhibitors used in clinical settings, PD-1 (nivolumab, pembrolizumab), PD-L1 (atezolizumab, durvalumab) and CTLA-4 (ipilimumab) are the most commonly targeted molecules [1]. CTLA-4 is a surface protein responsible for inhibiting TCR-mediated (T-cell receptor) activation of lymphocytes. Interaction of CTLA-4 with B7 family ligands counteracts their binding to co-stimulatory protein CD28 [4]. In contrast, the signalling pathway involving PD-1 and its ligands PD-L1/2 is directly affecting TCR, but the most recent studies also highlight CD28 as the primary target of that axis [5,6]. Another member of the B7 protein family, B7-H3, is decreasing the reactivity of T cells via the blocking of selected cytokines: IFN-γ, IL-2, IL-10, IL-13 [7].

Due to the presence of, inter alia, high levels of PD-1+ CD8+ T cells in the tumour microenvironment, especially at relapse after bone marrow transplant, AML is a potential candidate for therapeutic modulation of that immune checkpoint molecule [8,9]. In mice models, blockage of PD-1/PD-L1 showed substantial efficacy in improving the expansion and anti-tumour activity of cytotoxic T cells at the cancer cell site [10]. The most recent in vitro study on leukemic cells and lenalidomide in combination with nivolumab (anti-PD-1), suggested that certain populations of cells might benefit more from ICI use. That combination was shown to be highly effective in supporting a population of leukemic antigen-specific cytotoxic T cells, especially toward blasts of the NPM^mut^ AML group [11]. In contrast, blockage of PD-L1 on leukemic cells was associated with cell cycle arrest, inhibition of proliferation and activation of apoptotic pathways. These data supported the disadvantageous role of tumour-related PD-L1 in AML patients, in which high levels were associated with significantly worse survival [12]. Importantly, the poor prognosis was also reported in AML subjects with high PD-L1 within regulatory B cells, being possibly another target for specific therapy [13]. The protein of the B7 family, B7-H3, is another checkpoint with exclusively high expression in AML compared to other haematological malignancies and healthy controls [14].

Immunotherapy introducing blocking of the immune checkpoint proteins has proved its high potential in solid tumours. More recently, there are high expectations for transferring that knowledge into treatment of haematological malignancies. A cautious approach is necessary considering reported risk of toxicity or GvDH in those patients, depending on various factors including cancer type, immunological status and previous response of patients to the standard therapy [15]. However, other clinical studies consider a combination of blocking antibodies (nivolumab) with idarubicin and cytarabine as relatively safe and well-tolerated, and all potential adverse events possible to manage in most cases [16]. The satisfactory potential of the selected immune checkpoint inhibitors, namely anti-PD-L1 antibody (nivolumab), has been demonstrated so far in refractory or relapsed Hodgkin’s lymphoma. Importantly, Reed–Sternberg cells present in that malignancy shows an elevated expression of both, PD-L1 and PD-L2 proteins on their surface [17].

Previous reports suggest a combination of the standard chemotherapy with inhibitors of the checkpoint molecules. In animal models, application of gemcitabine with anti-CTLA-4 antibody showed improved survival compared to the monotherapy with each drug separately, predominantly through the induction of anti-tumour immune responses [18]. In contrast, a therapeutic approach involving azacytidine and durvalumab (blockers of the PD-L1 interaction with PD-1) showed no improvement in older AML patients’ survival, aged above 65 years [19]. Another clinical trial demonstrated the same efficacy of the other PD-L1 blocker, avelumab, even in younger subjects [20]. Data obtained from solid tumours treated with immune checkpoint inhibitors (ICIs) might suggest that a lower occurrence of immune-related adverse events (irAE) itself could be a sign of weak response to those drugs [21]. Thus, proper selection is required for applying ICIs to those patients that might actually benefit. Importantly, less than one-third of patients undergoing ICI implementation achieve therapeutic success from that form of treatment [22]. Subjects with tumour cells expressing high levels of PD-L1 were responding more efficiently to the blocking of its receptor—PD-1 (pembrolizumab) [23]. Co-expression of PD-1 with CTLA-4 on tumour-infiltrating lymphocytes (TILs) has also been linked to a better survival of patients [24]. The number of novel biomarkers for the personalization of the immune checkpoint inhibitor therapy is constantly growing. Besides approved tests like immunohistochemistry of PD-L1, tumour mutational burden (TMB) or microsatellite instability (MSI), experimental tests are gradually verified in predicting cancer therapy outcomes, namely immunoscore, TCR diversity, circulating immune-related proteins, and microbiome [25].

Additional, equally important, aspects that should be considered when estimating the potential efficacy of ICIs should be the level of their targets. In accordance, here we assessed expression patterns of most therapeutically promising checkpoint molecules, PD-1, PD-L1, CTLA-4, and B7-H3, in AML patients. The above-mentioned studies suggested the significance of the proper selection of the patients that might benefit from the blocking of those proteins. Therefore, we implemented clinical outcome-based stratification and molecular risks in analysing differences. Our study is the first one showing substantial changes in selected checkpoint molecules and their association with AML patients’ response to therapy and further survival.

## 2. Materials and Methods

### 2.1. Patients Characteristics

All experimental procedures were approved by the Bioethics Committee of Medical University of Bialystok, per institutional guidelines (R-I-002/393/2018). After obtaining informed consent, a total of 72 samples collected from previously untreated acute myeloblastic leukaemia were included in the study. The AML diagnosis was established according to the WHO criteria [26]. The study did not include patients with APL (acute promyelocytic leukaemia) due to different biology, treatments and outcomes. Other active malignancies and concomitant uncontrolled infections were the main exclusion criteria. 

The treatment responses and cytogenetic risk were evaluated according to the 2017 ELN [27]. Molecular testing included screening for mutations in *FLT3* (both for internal tandem duplications (ITDs) and tyrosine kinase domain (TKD) mutations at codons D835 and I836), *NPM1*, *CEBPA*, and *RUNX1* genes. Due to the lack of access, none of the patients received an inhibitor despite the FLT3 mutation. 

All patients included in the observation received induction therapy according to one of the protocols, i.e., the “3+7” scheme or DAC, according to the department standard [28]. Briefly, Daunorubicin 60 mg/m^2^ 30 min iv infusion and Cytarabine 200 mg/m^2^ 12 h iv infusion, 2 h infusion, on days 1–7 for the 3+7 scheme and additionally for patients receiving the DAC regimen, Cladribine 5 mg/m^2^ 2 h iv infusion on days 1–5. All patients who achieved complete remission (CR) received 2 or 3 cycles of consolidation treatment based on high doses of Cytarabine. Then, in the absence of contraindications, an allogeneic hematopoietic stem-cell transplantation (allo-HCT) was offered as soon as matched donors were available. 

Patient characteristics are listed in Table 1 in the main manuscript body. Their median age at the time of sample collection was 54, and the range was 18–67. Thirty-five subjects were female, and 37 were male. Patients were classified according to the ELN 2017 as follows: 8 patients had good risk (3 patients with *t* (8;21), 3 with *inv*(16), one with biallelic mutated core-binding factor leukaemia (CEBPA*_mut_*), one mutated nucleophosmin (NPM1*_mut_*) without FLT3-ITD), 42 subjects had intermediate risk (including diploid karyotype features with five both mutated nucleophosmin (NPM1*_mut_*) and FLT3-ITD^high^, and 6 with FLT3-ITD^low^ without NPM1*_mut_*), 22 subjects were classified as the unfavourable risk group (with del(5q), del(7q) and 5 patients with FLT3-ITD^high^ and wild-type NPM1 or complex (≥3) abnormalities). Of the enrolled patients, 60 (83%) received the DAC regimen and 12 (27%) the “3+7” regimen. The efficacy of the induction therapy was established according to the International Working Group criteria; [29] among the selected samples, 63% of patients achieved CR after the first induction and 37% were non-responders (NRs). One patient died during the induction. The patients who did not respond were given salvage therapy according to the standards of the haematology clinic. Intermediate-risk patients with a sibling donor and high-risk patients with a matched donor were offered an all-HSCT as soon as matched donors were available.

### 2.2. Study Material

Freshly obtained EDTA anticoagulated whole venous blood was processed according to the standard operating procedures of the Medical University of Bialystok Biobank. Briefly, blood samples were centrifuged at room temperature for 5 min 400× *g*, followed by plasma separation. Collected plasma was centrifuged at 4 °C for 5 min at 1200× *g* to remove residual cells. Aliquoted biofluids were biobanked in a −80 °C controlled environment. The cell pellet was refiled with PBS (Corning, Glendale, CA, USA) and placed on density gradient (Histopaq-1077; Merk, Darmstadt, Germany). The specimens were centrifuged for 25 min at room temperature. Finally, the interphase was collected and washed in PBS for 5 min at 4 °C. Freshly isolated PBMCs were counted, cryopreserved, and stored in LN2. Before FACS staining, cryopreserved PBMCs were rapidly thawed in a water bath at 37 °C and immediately resuspended in cell culture medium (RPMI-1640; Thermo Fisher, Waltham, MA, USA) supplemented with 10% FBS; PAA. The cells were centrifuged (at 300× *g*, room temperature) to wash off the cryoprotectant. Next, the cell pellet was resuspended with fresh medium suspended in the pre-warmed fresh cell culture medium and plated in 24-well polystyrene culture plates, followed by incubation for 3 h in normal conditions (37 °C, 5% CO_2_ saturation). Finally, the cells were harvested and counted in a Burker chamber using trypan blue (Thermo Fisher, Waltham, MA, USA). For the flow cytometric staining, at least 10 × 10^6^ of the isolated cells with confirmed viability were used.

### 2.3. Flow Cytometry

As previously described, a number of PD-1-, PD-L1-, CTLA-4-, and B7-H3-expressing CD4+ T cells and AML cells were assessed through flow cytometry (FACSCanto II, Becton Dickinson, San Jose, CA, USA). The cells were washed, resuspended in PBS (Corning, Glendale, CA, USA), and stained with Zombie Viability Dye (Biolegend, San Diego, CA, USA), according to the manufacturer’s instructions. The specimens were washed and resuspended in 200 μL of staining buffer (PBS with EDTA (Thermo Fisher) and 5% bovine albumin (Sigma-Aldrich)). The cells were stained with a panel of fluorochrome-conjugated monoclonal antibodies anti-CD3, anti-CD4, anti-CD33, anti-CTLA-4, anti-PD-1, anti-PD-L1, anti-B7-H3 (Appendix A), for 30 min at room temperature in the dark. All used antibodies were titrated prior to use. Finally, the cells were washed and fixed with Cell Fix (Becton Dickinson, San Jose, CA, USA). Appropriate fluorescence minus one (FMO) controls were applied to set proper gating. The representative gating strategy is presented in Appendix A. Initial gating was based on the morphology of PBMC, followed by the selection of single cells and exclusion of dead cells. Presence of CD3 and CD33 allowed for gating of T lymphocytes (CD3+CD33-) and blast cells (CD3-CD33+). Additional CD4 marker was used to distinguish the population of helper T cells (Th). Selected immune checkpoint proteins were analysed in context of their frequency within tested populations and surface expression (mean fluorescence intensity, MFI). The data were analysed with FlowJo (Becton Dickinson, San Jose, CA, USA).

### 2.4. Biostatistical Analysis

Statistical analysis was performed using GraphPad Prism 8 Software (GraphPad Software, San Diego, CA, USA). The Mann–Whitney U test was implemented considering a non-normal distribution of the data. Within manuscript graphs, the data are presented as medians with interquartile range (25th to 75th percentile), including presentation of individual patients’ results. Kaplan–Meier estimates and the log-rank test was applied to determine overall survival differences between studied subgroups. The level of statistical significance was set at *p* < 0.05.

## 3. Results

### 3.1. Baseline Levels of Selected Immune Checkpoint Proteins within Th Lymphocytes and Blast Cells of Patients with AML

First, we aimed to analyse baseline values of the helper T cells expressing studied immune checkpoint proteins between AML patients divided on the basis of their treatment responses. In accordance, treated subjects were divided into complete responders (CRs) and non-responders (NRs). Leukaemic patients from the NR group demonstrated significantly higher frequencies of Th cells expressing PD-1 compared to CR subjects (*p* = 0.0015). We did not show any significant variations between CR and NR patients in percentage of PD-L1-, CTLA-4- or B7-H3-positive Th cells. In addition, no differences in the surface mean expression of the remaining immune checkpoint proteins were observed (Figure 1A,B).

In contrast to lymphocytes, blast cells did not show any difference between CR and NR patients with AML; in the context of PD-1, both in the percentage of the cells and surface expression of the protein. The only studied marker demonstrating variations in expression on blast cells was CTLA-4. Patients of the NR group had a significantly higher percentage of CTLA-4-positive leukaemic cells versus CR patients (*p* = 0.0384). Similarly, elevated mean surface expression of CTLA-4 was also observed in the NR subjects (*p* = 0.0275). Tested blast cells did not show any statistically significant differences between CR and NR in reference to the remaining immune checkpoint proteins (Figure 2A,B).

### 3.2. Expression of Selected Immune Checkpoint Proteins within Th Lymphocytes and Blast Cells of AML Patients Stratified on the Basis of Molecular Cytogenetic Risk

In the subsequent step, we evaluated differences in selected immune checkpoint proteins expression between different risk groups of AML patients. Accordingly, molecular cytogenetic risk assessment was based on the European LeukemiaNet 2013 (ELN) recommendations. That allowed as to distinguish patients with favourable (1), intermediate (2), and adverse (3) prognosis. Regarding helper T lymphocytes, we found that the percentage of those cells expressing PD-L1 was significantly higher in the intermediate-risk group versus patients in the favourable group (*p* = 0.0013). Subjects from the intermediate group also showed elevated mean surface expression of PD-L1 compared to both favourable (*p* = 0.0003) or adverse (*p* = 0.0015) prognosis. None of other immune checkpoint molecules showed significant variations in reference to the percentage of expressing Th cells. Further analysis of surface levels of the proteins revealed significantly reduced expression of CTLA-4 in intermediate prognosis AML patients when compared to the favourable (*p* = 0.0348) or adverse (*p* = 0.0227) group (Figure 3A,B).

Interestingly, blast cells of AML patients did not demonstrate any significant differences between risk groups in reference to cells with expression of immune checkpoint proteins. That refers to both percentage of cells and mean surface expression of selected immune checkpoint proteins within blast cells. Nevertheless, despite a lack of statistical significance, median levels of PD-1- and PD-L1-positive leukemic cells of intermediate and adverse prognosis groups showed reduced values (Figure 4A,B). 

### 3.3. Influence of Selected Immune Checkpoint Proteins Levels within Th Lymphocytes and Blast Cells on the Survival of the AML Patients

Considering the above information as the next step, we wished to analyse whether cells expressing selected immune checkpoint proteins, Th lymphocytes and blast cells, have influence on the survival of the AML patients. Therefore, we divided studied subjects into groups with lower and higher levels of the cells with specific proteins, on the basis of the median values of the parameters. First, we demonstrated that patients with a lower frequency of PD-1+ cells within Th lymphocytes had a tendency for better survival rates compared to those with high levels (*p* = 0.0843). A similar trend seemed to be present in the context of CTLA-4-positive Th cells; however, no statistical significance was achieved. Noteworthily, we found that AML patients who demonstrated a higher frequency of Th lymphocytes expressing B7-H3 had significantly higher survival rates compared to lower-value subjects (*p* = 0.0471). In observed AML patients, those with a lower percentage of B7-H3 Th cells maintained a probability of survival of around 35% at the sixth year of monitoring, whereas higher levels subjects reached 0% earlier after approximately 4.5 years. Levels of mean surface expression of selected immune checkpoint proteins on Th cells did not show influence on the survival of studied AML patients (Figure 5A,B).

In reference to blast cells, we did not show any influence of the selected immune checkpoints levels on the survival of the AML patients. Interestingly, although not statistically significant, a higher frequency of leukemic cells expressing PD-1 seemed to lead to an extended survival of patients. Those subjects after 8 years of monitoring showed a probability of survival close to 40%, and lower-level AML patients demonstrated 0% early survival after the fourth year of observation. Chances of patients’ survival seemed also to be slightly higher in context of surface expression of PD-1 on blast cells; however, that was observed only around the 12th to 24th month of therapy. Despite a lack of statistical significance, a trend for the better survival of leukemic cells with lower B7-H3+ frequency was reported higher within 1st to 2nd year of monitoring. Within the next months, those differences were diminished (Figure 6A,B).

## 4. Discussion

Application of immune checkpoint inhibitors dominates the area of solid tumour therapy, with increasing numbers of novel antibodies approved for clinical use. Nonetheless, the implementation of nivolumab in Hodgkin lymphoma sheds a light on new opportunities in the management of haematological malignancies [1]. It is well documented that, in general, the extensive activation of immune cells, also in the course of AML progression, can be associated with exhaustion of T lymphocytes and an increase in inhibitory molecules like PD-1. Interestingly, lack of PD-1 in the knock-out AML mouse model led to improved survival of the animals, thus indicating a crucial contribution of that protein in the disease pathogenesis [10]. Noteworthily, the PD-1/PD-L1 axis is just one of numerous immune checkpoints currently being tested as potential targets for novel immunotherapies of AML [3].

Increased expression of the PD-1 receptor and its importance as a prediction and prognostic factor has been demonstrated in various solid malignancies, including lung, stomach, breast, and skin tumours. Interestingly, elevated expression of PD-1 was observed in both tumour cells and immune cells (tumour-infiltrating and -circulating cells). Its elevated expression has been shown to be associated with a more aggressive course of the disease, higher numbers of local and distal metastases, and shortened disease-free survival and overall survival [30,31]. We found that patients with AML not responding to the DAC/DA therapeutic protocols (NR) demonstrated significantly higher levels of helper T lymphocytes expressing PD-1. As suggested in previous study, that fact can be associated inter alia with exhaustion of the T cells. Concomitantly to our observations, Zhou et al. demonstrated an association between higher frequencies of CD25^low^PD-1+ CD8+ T cells and AML progression. That PD-1-expressing population of peripheral blood lymphocytes was significantly lower in subjects with a complete remission to the therapy [32]. Similarly, PD-1+TIM-3+ effector T cells from bone marrow were also indication of poor therapeutic response, with higher frequency in non-responders [16]. In addition, a combination of PD-1^high^ and TIM-3+ markers on T cells revealed that high levels of that subset were also associated with higher risk of AML relapse after transplantation of allo-SCT [33], due to the clear stratification of patients into CR and NR groups, we managed to highlight the role of PD-1 level within Th lymphocytes in AML subjects’ response to the therapy. An important novelty of our study data is the fact that high values of PD-1 only within T lymphocytes were sufficient for determining non-responding AML patients. Most recent studies, like the one mentioned above or the study of Noviello et al., suggested a combination of several exhaustion markers in establishing the possibility of disease relapse: PD-1+, Eomes+ and T-bet+ Memory Stem T cells infiltrating bone marrow. The same study indicated, concomitantly, that PD-1-related variations between relapsing patients or those with complete remission might be closely associated with the phenotype of the T cells, as higher levels of PD-1 in subjects with unfavourable events were mostly limited to effector memory and memory stem T cells, as far as CD8+ lymphocytes are concerned [34]. A summary of the most recent data on PD-1 expression within immune cells might suggest the possible use of the protein as a target for immunotherapeutic approaches. Considering the indicated exhaustion of the CD8+ T cells, high levels of PD-1-positive cells might be a contradiction to blockage with rhPD-L1 molecules or anti-PD-1 antibodies [32]. That fact is associated with the resistance of those terminally differentiated immune cells to the specific inhibition, and results in poor clinical outcomes [35]. Taken together, those reports might partially explain the unsatisfactory efficiency of the PD-1-PD-L1 axis blockage in the last years, using, for example avelumab (anti-PD-L1 antibody) with standard chemotherapy [36]. On the other hand, it was also found that a dysfunction of T cells can be restored in vitro with the inhibition of anti-PD-1. That, however, requires further investigation to assess the possibility of transferring those results into a clinical setting [9]. Interestingly, exhausted T cells infiltrating bone marrow with at least one of the exhaustion markers (PD-1, Tim-3 or 2B4) showed high anti-leukemic specificity, compared to lymphocytes with no presence of those markers. That was based on the higher production of Granzyme B and the elimination of autologous blast cells in cell culture [34]. Yet, despite those in vitro-reported revelations, here we demonstrated that at least a lower frequency of PD-1-positive Th cells is rather associated with a non-significant trend for a better chance of the AML patients’ survival. Including high levels of Th cells with PD-1 expression in AML non-responders, that group of patients might possibly benefit most from blocking the PD-1/PD-L1 pathway. In addition, improved survival of subjects with lower PD-1 expression on bone marrow mononuclear cells was also observed in myelodysplastic syndrome (MDS) [37].

Unlike the PD-1 receptor, the predictive and prognostic value of PD-L1 protein expression varies according to the type of cancer. PD-L1 overexpression on tumour cells strongly correlates with weaker treatment outcomes and adverse prognosis in gastric cancer, hepatocellular carcinoma, renal cell carcinoma, oesophageal cancer, prostate cancer, and ovarian cancer [38]. However, opposite results were observed in breast cancer and Merkel cell carcinoma [39]. In AML, the role of PD-L1 remains elusive. In the previous studies, AML patients were shown to demonstrate a higher expression of PD-L1 within bone marrow cells [12]. Subsequent results of our study did not report any initial differences when patients were stratified into CR and NR groups, neither in leukocytes nor blast cells of peripheral blood. Assessment of PD-L1 revealed that higher levels within Th lymphocytes are present in AML patients with intermediate prognosis in the context of cytogenetic molecular risk assessment. Concomitantly, however, we did not find any influence of that protein expression on the survival of AML subjects. That fact might reduce the significance of the potential uses of immune checkpoint inhibitors targeted at PD-L1 in that haematological condition. On the other hand, a proper selection of subjects with that molecule might potentially improve response to immunotherapy. In the pembrolizumab therapy of myeloid malignancies relapsing after allo-HCT, subjects with a high expression of PD-L1 within lymphoma cells showed a substantial complete response to therapy [40]. That might be related to the predominance of CD19+ cells that, together with regulatory B cells (Breg), were shown as a great source of PD-L1 in AML [13]. In AML subjects aged above 65 years, there were no improvements in the chemotherapy (azacytidine) outcomes when blocking of PD-L1 with PD-1 was introduced using durvalumab. On the other hand, in that study group, immune checkpoint inhibition was not associated with the hazardous side effects reported previously, with safety comparable to that of azacytidine/decitabine/cytarabine with venetoclax [19]. A similar safety of anti-PD-L1 with still a lack of favourable clinical improvement of the standard therapy was reported by Saxena et al. in their clinical trial with azacytidine and avelumab. Interestingly, the same group found significantly higher levels of PD-L2 compared to PD-L1 in both bone marrow and the peripheral blood blasts of AML subjects [20]. These data, together with the results presented here, might suggest the need for focusing attention on the PD-L2 molecule in the context of its contribution in AML pathogenesis, and even therapeutic applications. Another option is an in-depth insight into additional pathways that could influence the response of leukemic cells to therapy with anti-PD-L1. Soltani et al. suggested that certain metabolic pathways of leukemic cells that were affected by blockers of PD-L1, might be promising targets for novel drugs implemented additionally in ICI therapy [41].

Despite a lack of statistical significance, we observed in another research group study substantially higher levels of CTLA-4 within CD3+ lymphocytes in AML patients experiencing post-transplant (HSCT) relapse. Interestingly, their results did not reveal any initial differences between CR and NR subjects in the context of CTLA-4 or PD-1 within both helper CD4+ and cytotoxic CD8+ T cells [34]. Here, we essentially broaden our knowledge on the potential of CTLA-4 as a predictor. Levels of CTLA-4 on blast cells were distinguished in the dominant population allowing for the differentiation between AML patients with complete remission (low levels) and non-responders (high level). Despite no expression or frequency changes in CTLA-4 within Th cells between CR and NR, we demonstrated that lymphocytes demonstrating a surface presence of that immune checkpoint protein belonged to the group with intermediate prognosis—based on the cytogenetic molecular assessment. Whether targeting CTLA-4 can improve outcome of the AML subjects’ therapy still remains one of the topics requiring further research and clinical trials. Despite the occurrence of cytotoxicity and GvHD in some patients with relapsed haematological cancers, treatment with ipilimumab was able to achieve a complete response among the selected subjects. Extramedullary AML patients relapsing after allo-HSCT, were shown to present a favourable therapeutic response which sustained for more than a year after therapy [42]. In accordance, it seems that type of haematological neoplasm and even clinical situation of the patients critically affects the response of the implemented management. Considering our results, the expression of CTLA-4 on T lymphocytes in AML is another factor worth assessment prior to immune checkpoint blockage. We showed that those belonging to the group with intermediate prognosis (based on ELN recommendations) demonstrated a reduced expression of CTLA-4. Thus, we might presume a worse efficiency of inhibitors like anti-CTLA-4 recombinant antibodies in that case. It is worth mentioning that the modulation of different targets, like PD-1 with nivolumab, was shown to also affect other molecules. Higher doses of anti-PD-1 antibodies were found to increase the frequency of bone marrow CTLA-4+ T cells in the non-responders’ group of AML patients [43]. A single dose of ipilimumab increased the infiltration of CD8+ lymphocytes and led to a higher perforin expression in sites of leukaemia cutis [42]. An induced expansion of CD8+ T lymphocytes specific towards tumour cells was also demonstrated when anti-PD-1 or anti-PD-L1 antibodies were used in AML patients [44]. Those observations were first made analysing responses of mice AML models to PD-L1 blocking [10]. A significant efficiency of anti-PD-1 antibodies (nivolumab) in combination with azacytidine was demonstrated in specific groups of patients with relapsed/refractory AML: HMA-naïve, Salvage 1 and high pre-treatment infiltration of the bone marrow with CD3+ lymphocytes (CD8 predominantly). Interestingly, that therapy also induced a higher expression of CTLA-4-positive T cells among NR patients, and that justifies current trials with combinatory management with anti-PD-1 and anti-CTLA-4 [43].

In contrast to above mentioned ICIs, the B7-H3 molecule is a relatively newly discovered checkpoint molecule. It is commonly acknowledged as a promising target in cancer immunotherapy. B7-H3 is overexpressed in several solid malignancies, including non-small-cell lung cancer, prostate cancer, and breast cancer. Usually, its presence was associated with high aggressiveness, poor treatment outcomes, and decreased survival rates [7]. The role of the B7-H3 receptor in AML still needs to be fully elucidated. To date, total mononuclear cells of bone marrow and peripheral blood of AML were shown before to demonstrate the elevated expression of B7-H3 compared to the healthy control group [45]. In the current study, we did not find any significant differences in of B7-H3 levels within blast cells or lymphocytes, in the context of response to therapy or cytogenetic molecular risk in leukemic samples. Noteworthily, however, a significantly higher survival of AML patients was demonstrated in groups with a lower percentage of Th lymphocytes expressing B7-H3. These clinical data substantially complement previous results of available dataset analysis or clinical observations that highlighted a poor prognosis for survival of AML subjects showing a high expression of B7-H3 [45,46]. Although statistically significant, results of our survival analysis must be used with caution when assessing other patients’ risk due to the fact that borderline *p* levels were reached. The molecule presence was additionally revealed to be associated with a doubled risk of unfavourable survival in the whole group of acute leukemic patients (ALL and AML) [47]. Importantly, apart from being a significant contribution to the survival of the patients, B7-H3 was found to be positively correlated with CD80 and CD86 that interact with the CTLA-4 molecule [14]. Monitoring of B7-H3 could be a useful predictor of the AML management outcome, and potentially a parameter for determining a therapeutic approach after further determination of populations associated with that molecule. In addition, therapeutic application could also be considered as B7-H3 was detected on AML subjects’ NK cells. Targeting the population of those cells with a high potential in antibody-dependent cell-mediated cytotoxicity (ADCC) could be beneficial, as blocking of the immune checkpoint molecule significantly prolonged survival in patient-derived xenograft (PDX) models [45].

## 5. Conclusions

Here, we showed that AML patients not responding to the induction therapy have higher levels of helper T cells expressing PD-1 compared to those with complete remission. These data translate into subsequent prognosis for leukemic patients, with elevated PD-1-positive cells, indicating a risk of reduced survival. No significant changes in PD-L1 might diminish the potential role of that molecule as a target for immune checkpoint inhibitors. Eventually, more attention should focus on other pathways related to PD-L1 activity, or PD-L2 which dominates in AML cells. Furthermore, we showed that the assessment of the frequency of T helper cells expressing B7-H3 might serve as another prognostic marker for overall survival in AML patients. Despite substantial results, we are aware of certain limitations of the study. Those include, inter alia, the focusing of our attention on selected subset of lymphocytes, helper T cells, as key regulators of adaptive immune responses. In addition, studied patients were treated long before the introduction of the 2022 ELN risk categorization. Therefore, here, we implemented an evaluation in accordance with 2017 ELN.

Numerous recent papers did not report the expected high efficiency of the immune checkpoint inhibitors in the therapy of the AML. Still, however, most of these studies do not suggest abandoning the topic, as the potential of that immunotherapy seems to be hidden within proper protocol establishment [19,20]. Some researchers indicated that the application of these blockers might be even essential in preventing post-transplant relapses in AML. That might require a highly specific combination allowing for the restoration of immune cells reactive towards neoplastic cells without potentially exacerbating GvHD [34,44]. The results of our study highlight the significance of the selected immune checkpoint proteins and their potential in predicting treatment outcomes and survival of patients with acute myeloid leukaemia. Nonetheless, further studies are needed to better understand the exact mechanisms associated with these molecules’ activity in AML, and to validate the proposed biomarker candidates on greater multicentre cohorts.

## Figures and Tables

**Figure 1 cancers-15-04487-f001:**
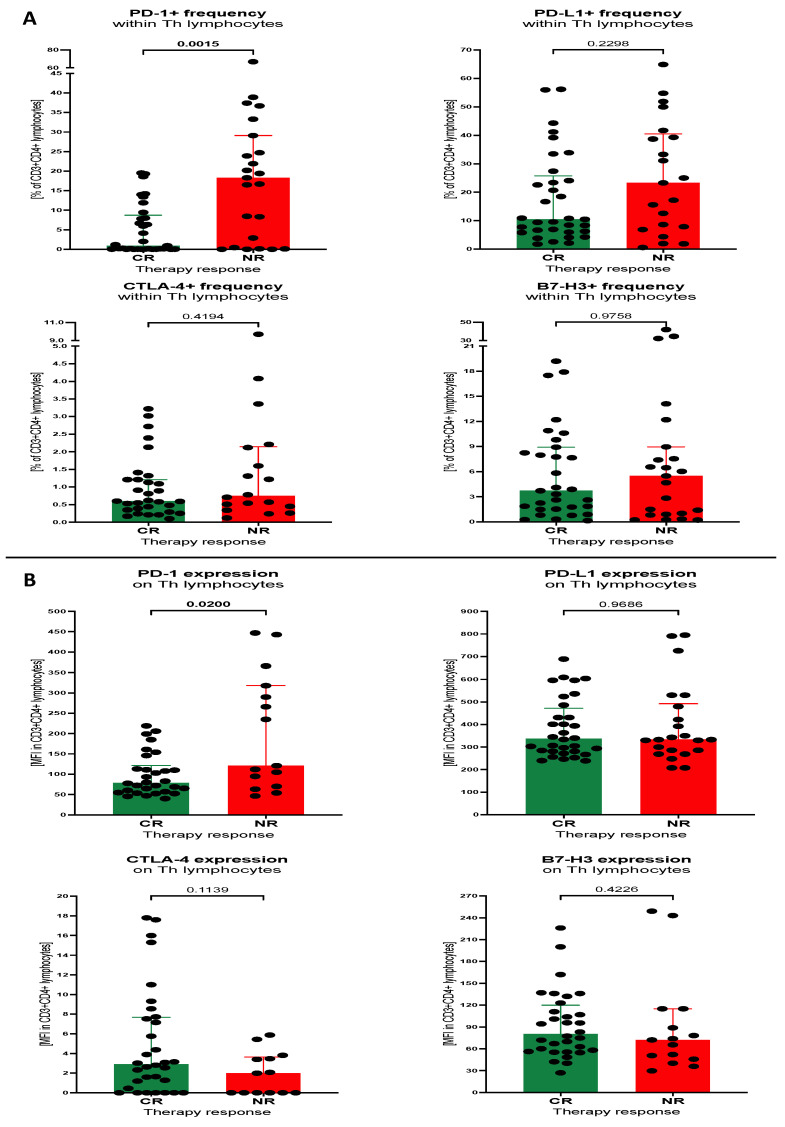
Levels of selected immune checkpoint proteins on Th lymphocytes of AML patients. Immune checkpoint proteins PD-1, PD-L1, CTLA-4 and B7-H3 were analysed between CR and NR subjects within Th lymphocytes in context of percentage of positive cells (**A**) and mean fluorescence intensity (surface expression) of selected markers (**B**). Data presented as median values with interquartile range. Exact *p* values are provided within graphs, and statistically significant differences indicated with bolding for *p* < 0.05.

**Figure 2 cancers-15-04487-f002:**
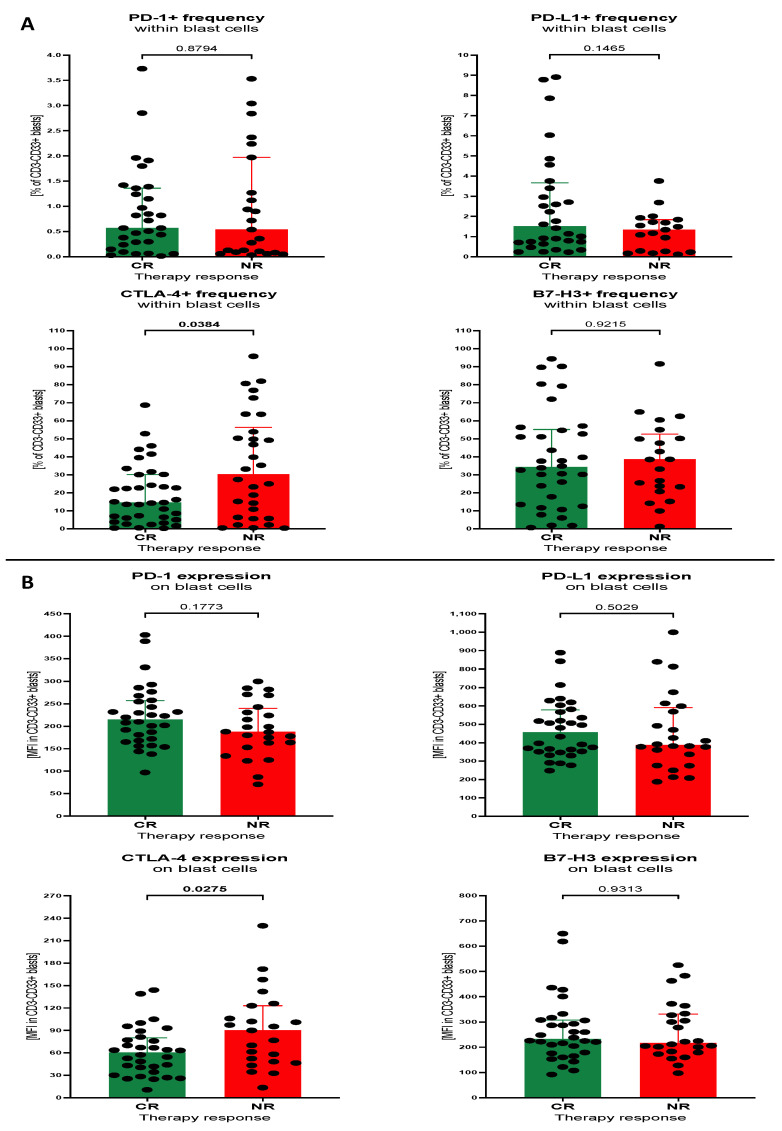
Levels of selected immune checkpoint proteins on blast cells of AML patients. Immune checkpoint proteins PD-1, PD-L1, CTLA-4 and B7-H3 were analysed between CR and NR subjects within blast cells, in context of percentage of positive cells (**A**) and mean fluorescence intensity (surface expression) of selected markers (**B**). Data presented as median values with interquartile range. Exact *p* values are provided within graphs, and statistically significant differences indicated with bolding for *p* < 0.05.

**Figure 3 cancers-15-04487-f003:**
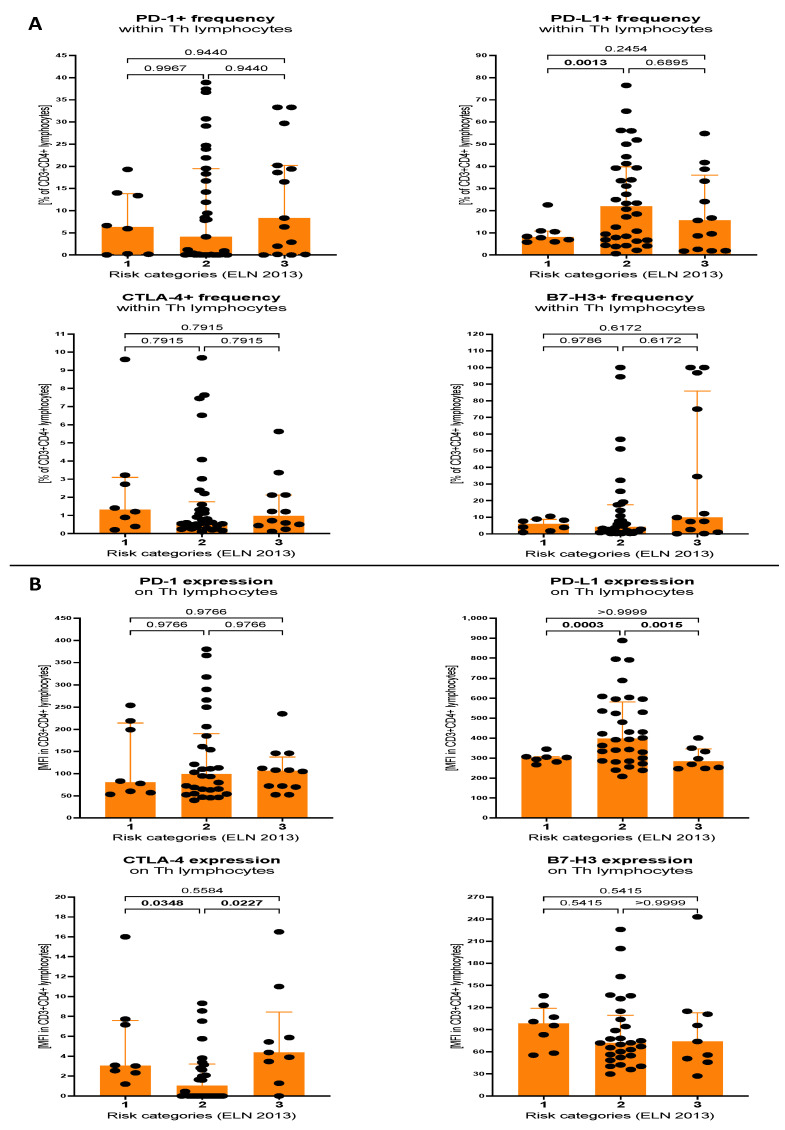
Levels of selected immune checkpoint proteins within Th lymphocytes of AML patients, stratified on the basis of ELN 2013 risk categories. Differences were analysed between AML patients with favourable (1), intermediate (2) and adverse (3) prognosis. Immune checkpoint proteins—PD-1, PD-L1, CTLA-4 and B7-H3 were analysed within Th lymphocytes, in context of percentage of positive cells (**A**) and mean fluorescence intensity (surface expression) of selected markers (**B**). Data presented as median values with interquartile range. Exact *p* values are provided within graphs, and statistically significant differences indicated with bolding for *p* < 0.05.

**Figure 4 cancers-15-04487-f004:**
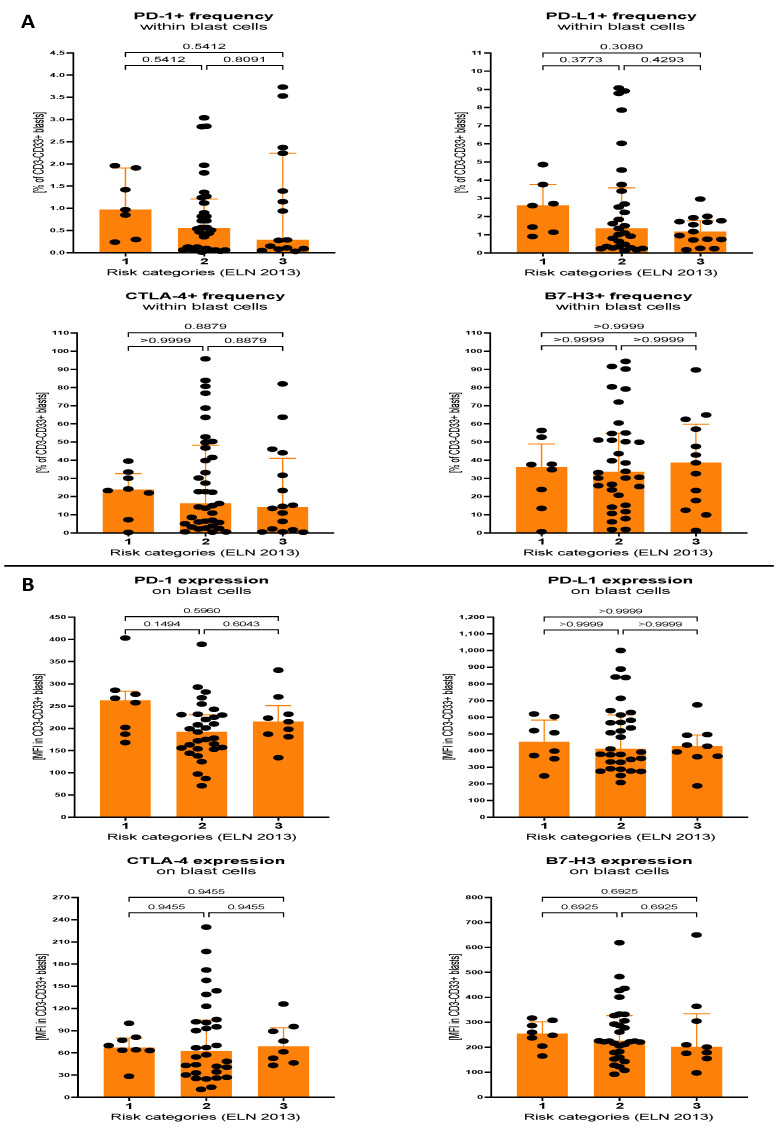
Levels of selected immune checkpoint proteins within blast cells of AML patients, stratified on the basis of ELN 2013 risk categories. Differences were analysed between AML patients with favourable (1), intermediate (2) and adverse (3) prognosis. Immune checkpoint proteins PD-1, PD-L1, CTLA-4 and B7-H3 were analysed within blast cells, in context of percentage of positive cells (**A**) and mean fluorescence intensity (surface expression) of selected markers (**B**). Data presented as median values with interquartile range. Exact *p* values are provided within graphs, and statistically significant differences indicated with bolding for *p* < 0.05.

**Figure 5 cancers-15-04487-f005:**
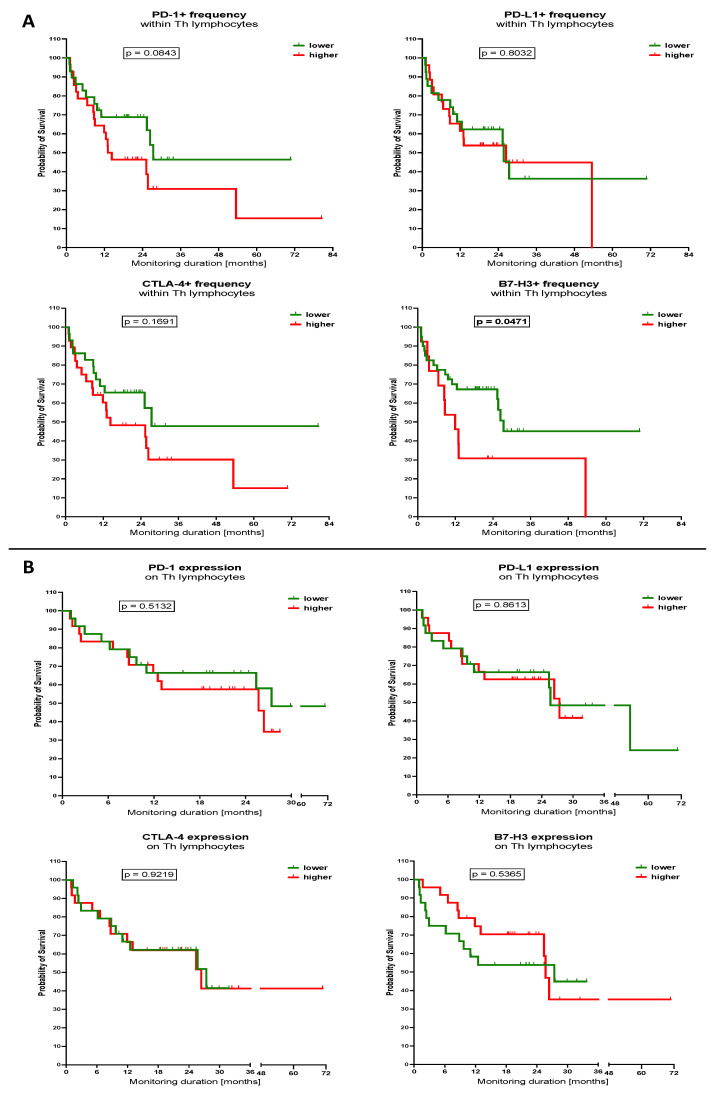
Influence of selected immune checkpoint proteins within Th lymphocytes on the AML patients’ survival. Kaplan–Meier curves demonstrated survival probability of AML patients in context of different levels (lower/higher, based on the median value) of immune checkpoint proteins PD-1, PD-L1, CTLA-4, and B7-H3 in context of percentage of positive cells (**A**) and mean fluorescence intensity (surface expression) of selected markers (**B**) in Th lymphocytes. Exact *p* values are provided within graphs, and statistically significant differences indicated with bolding for *p* < 0.05.

**Figure 6 cancers-15-04487-f006:**
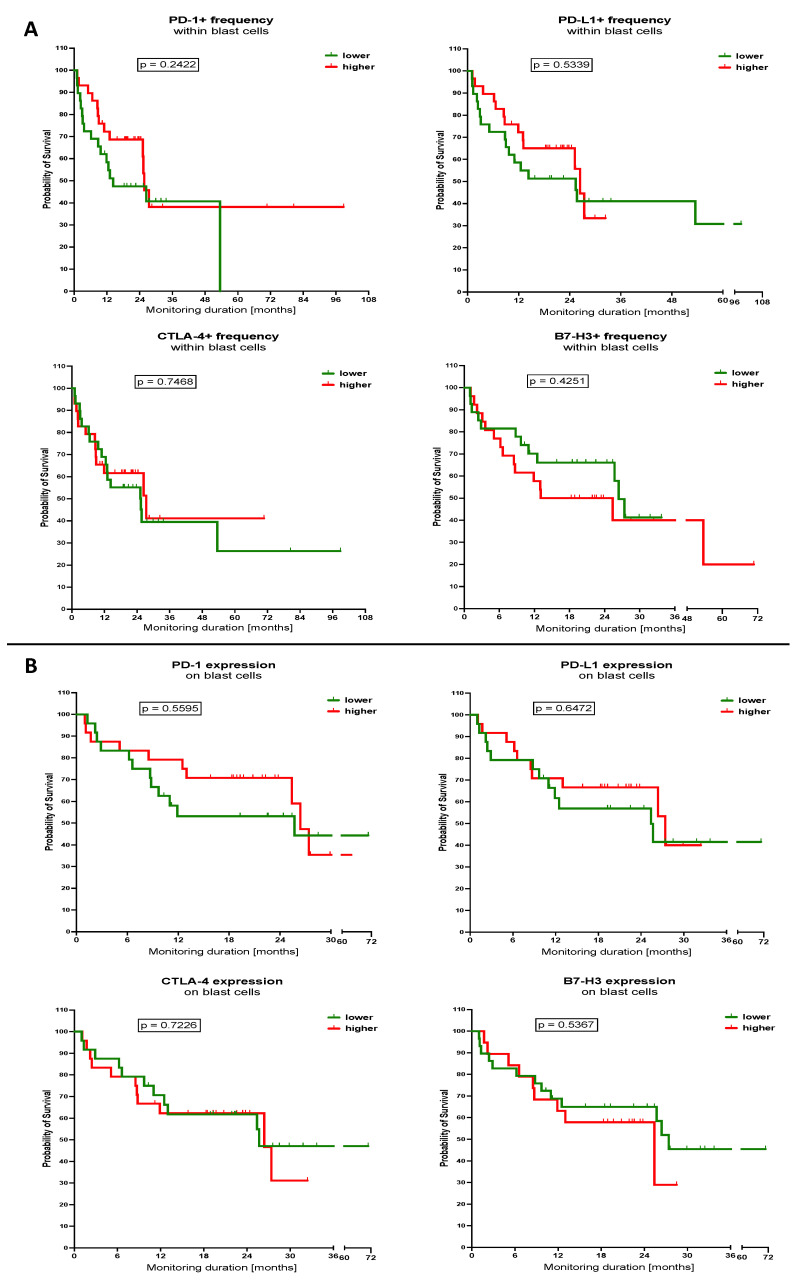
Influence of selected immune checkpoint proteins within blast cells on the AML patients’ survival. Kaplan–Meier curves demonstrated survival probability of AML patients in context of different levels (lower/higher, based on the median value) of immune checkpoint proteins PD-1, PD-L1, CTLA-4, and B7-H3, in context of percentage of positive cells (**A**) and mean fluorescence intensity (surface expression) of selected markers (**B**) in blast cells. Exact *p* values are provided within graphs, and statistically significant differences indicated with bolding for *p* < 0.05.

**Table 1 cancers-15-04487-t001:** Characteristics of the AML patients (*AML*—acute myeloid leukaemia; *CR*—complete remission; *NPM1_mut_*—mutated nucleophosmin; biallelic *CEBPA_mut_*—mutated core-binding factor leukaemia; *FLT3-ITD*—internal tandem duplication of Fms-like tyrosine kinase 3).

Number of Patients	72
Mean (range) age, year	54 (18–67)
Mean (±SD) white blood cell count (G/l)	38.1 (2.0–212.3)
Mean (range) percentage of blast cells in peripheral blood	47 (0–98)
Mean (range) of blastic cell percentage in bone marrow	56 (20–93)
**1. AML with recurrent genetic abnormalities**	*n* (%)
*t*(8;21) (q22;q22);(AML1/ETO)	3 (4%)
*inv*(16) (p13;q22) or *t*(16;16) (p13;q22); (CBFß/ MYH11)	3 (4%)
*Biallelic CEBPA_mut_*	1 (1%)
NPM1*_mut_* without FLT3-ITD	1 (1%)
*t*(9;11); MLLT3-MLL	2 (2%)
*NPM1*_wt_*FLT3*^low^/*NPM1*_mut_*FLT3*^high^	11 (15%)
*NPM1* _wt_ *FLT3* ^high^	5 (6.5%)
**2. AML with multilineage dysplasia without antecedent MDS**	6 (7.5%)
**3. AML therapy-related**	0 (0%)
**4. AML not otherwise categorized (FAB classification)**	n (%)
AML, minimally differentiated	6 (7.5%)
AML without maturation	10 (13%)
AML with maturation	15 (20%)
Acute myelomonocytic leukaemia (AMMoL)	7 (9%)
Acute monocytic leukaemia	8 (10%)
Induction therapy protocols: DAC/DA, *n*	60/12
CR achieved after the first induction, *n* (%)	45 (63%)
Favourable risk	8 (11%)
Intermediate I and II risk	42 (58%)
Unfavourable risk	22 (31%)

## Data Availability

The data that support the findings of this study are available from the corresponding author, upon reasonable request.

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
