# Peer review of "The Association between Immune Checkpoint Proteins and Therapy Outcomes in Acute Myeloid Leukaemia Patients"

_cancers, 2023, doi:10.3390/cancers15184487_

Round 1

Reviewer 1 Report

The study investigates the presence and expression level of particular immune checkpoint molecules in a local AML cohort and their potential impact on the prognosis and survival of the patients. Some comments are listed below:

(1) B7 family molecules include at least 10 members. Other than PD-1, PD-L1 and CTLA-4, I didn't see a specific reason/rationale for selectively choosing B7-H3 for analysis. Why is B7-H3 more important than other B7 members in AML?

(2) I believe personalized regimes in terms of the presence and expression level of the checkpoint molecules should be discussed more as it is the direction of most immunotherapies.

(3) Typos and grammar issues are seen throughout the manuscript. For example, in line 83-84, "graft vs. host disease" should be abbreviated as GvHD but not "GVDH" (in 434 and 501 as well), and "depending or" should be "depending on", etc.

Language should be further polished for obvious typos and grammar issues.

Author Response

We would like to thank the Reviewer for all the comments and suggestions regarding our manuscript. We believe that strict following of all the mentioned aspects allowed us to significantly improve the quality of the manuscript. Below we have attached responses to all the comments.

(1) B7 family molecules include at least 10 members. Other than PD-1, PD-L1 and CTLA-4, I didn't see a specific reason/rationale for selectively choosing B7-H3 for analysis. Why is B7-H3 more important than other B7 members in AML?

The selection of immune checkpoint proteins was based on the knowledge gaps requiring investigation in the context of their contribution to AML. In addition, we also wanted to focus on those proteins that have the highest potential for clinical implementation considering their use in other immunotherapies, like anti-CTLA-4, anti-PD-1 or antu-PD-L1 blocking antibodies. As a relatively newly discovered and already suggested as a promising target in other types of cancer, we decided to evaluate B7-H3 influence in the course of AML. Justification of the selected immune checkpoint proteins use is partially described within the Introduction and later extended in the Discussion section.

(2) I believe personalized regimes in terms of the presence and expression level of the checkpoint molecules should be discussed more as it is the direction of most immunotherapies.

In accordance with the Reviewer's suggestion, we extended the introduction with additional discussion on the personalization of therapy in the context of the implementation of immune checkpoint proteins.

(3) Typos and grammar issues are seen throughout the manuscript. For example, in line 83-84, "graft vs. host disease" should be abbreviated as GvHD but not "GVDH" (in 434 and 501 as well), and "depending or" should be "depending on", etc.

All mentioned typos and grammar errors have been corrected. English language specialists experienced with biomedical texts once again verified the whole manuscript.

Reviewer 2 Report

Bolkun et al. analyzed expression of the immune checkpoints PD-1, PD-L1, CTLA-4, B7-H3 in 72 AML patients. Results were correlated with response to therapy, ELN risk group and survival. The topic is interesting an immune checkpoints still hold a number of unanswered questions in AML. The manuscript is well written and the results well presented. The significance of results is moderate. The claims and conclusions are largely but not fully supported by the data. The discussion is well balanced.

Several major issues need to be considered:

1) The gating strategy for AML blasts solely relays on CD33 positivity. This is insufficient. Expression of CD33 is also found on maturating myeloid cells and on monocytes. In addition, in a few AML patients myeloid blasts can be negative for CD33. While the later might have been checked within study inclusion the former substantially limits the meaningfulness of results. Depending on the varying degree of remaining maturating myeloid/monocytic cells in the sample, results of marker expression on blasts may be skewed. I understand that the staining strategy cannot be adjusted retrospectively. Results for blasts showed rather be described as results for CD33+ cells throughout the figures and text and this limitation needs to be discussed in detail.

2) Survival data are not really convincing. 16 different variables are tested in Kaplan-Meier analysis. Results are not corrected for multiple testing (respectively no additional analysis multivariate analysis are performed) and only borderline levels of significance are reached. Keeping this in mind, results of survival data need to be discussed with caution. E.g. the abstract states “Elevated levels of PD-1 were associated with poorer patients’ survival.”, while this actually is a non-significant trend (p=0.08) despite the limitations mentioned above.

In addition, some minor point could be considered:

3) The gating strategy for T-cells via CD3 is ok but does not allow for analysis of T-cell subsets that might be more informative. This should be discussed as a limitation.

4) Both the WHO classification and the ELN risk categorization have been updated in 2022. It should be briefly discussed that not the latest versions of disease and risk classification have been used.

ok

Author Response

First, we would like to thank you for all the comments and suggestions. We believe that strict following of all the mentioned aspects allowed us to significantly improve the quality of the manuscript. Below we have attached responses to all the comments.

1) The gating strategy for AML blasts solely relays on CD33 positivity. This is insufficient. Expression of CD33 is also found on maturating myeloid cells and on monocytes. In addition, in a few AML patients’ myeloid blasts can be negative for CD33. While the later might have been checked within study inclusion the former substantially limits the meaningfulness of results. Depending on the varying degree of remaining maturating myeloid/monocytic cells in the sample, results of marker expression on blasts may be skewed. I understand that the staining strategy cannot be adjusted retrospectively. Results for blasts showed rather be described as results for CD33+ cells throughout the figures and text and this limitation needs to be discussed in detail.

We agree with the Reviewer that using CD33 as a marker of blast cells can have certain limitations. However, in our study, we based our gating strategy on the preceding diagnostic evaluation of the peripheral blood morphology and blast evaluation. In accordance, we included in the study those patients who were found to demonstrate the presence of CD33-positive blast cells (the flow cytometry when the diagnosis was made).  Regarding the presence of CD33 on other immune cells including monocytes, we decided to discuss that potential limitation within the manuscript. However, based on the recent data expression of CD33 on monocytes, if present, is very low and increases in hypoxic conditions, for example within tumor microenvironment (Olingy et al. Front Immunol. 2022).

2) Survival data are not really convincing. 16 different variables are tested in Kaplan-Meier analysis. Results are not corrected for multiple testing (respectively no additional analysis multivariate analysis are performed) and only borderline levels of significance are reached. Keeping this in mind, results of survival data need to be discussed with caution. E.g. the abstract states “Elevated levels of PD-1 were associated with poorer patients’ survival.”, while this actually is a non-significant trend (p=0.08) despite the limitations mentioned above.

The Reviewer rightly noticed that reported differences in survival considering selected parameters demonstrate rather borderline levels or even just non-significant trends. In accordance, to avoid incorrect conclusions we decided to clearly indicate the strength of the statistically significant differences. Also mentioned parts regarding PD-1 level influence on the survival rate were changed to highlight just only non-significant trend was observed in that case.

In addition, some minor point could be considered:

3) The gating strategy for T-cells via CD3 is ok but does not allow for analysis of T-cell subsets that might be more informative. This should be discussed as a limitation.

As indicated by the Reviewer, gating with CD3 only does not allow for investigation of changes within T cell subsets. In our gating strategy, we included an additional analysis of CD4, thus, we were able to monitor the population of CD3-CD4+ lymphocytes (helper T cells). We have corrected the Supplementary Figure 1 so the complete gating strategy is now presented. We must note here that we focused on the CD4+ T cells in our study as that population is considered most crucial in orchestrating adaptive immune responses.

4) Both the WHO classification and the ELN risk categorization have been updated in 2022. It should be briefly discussed that not the latest versions of disease and risk classification have been used.

We decided to include that crucial information within the Conclusions section when mentioning certain limitations of the study. There, we clearly indicated that studied patients were treated before the introduction of the new 2022 ELN categorization. Therefore, those subjects at the time of the study were still evaluated in the context of risk in accordance with 2017 ELN.

Round 2

Reviewer 1 Report

The authors have addressed the reviewer's comments properly.

Reviewer 2 Report

My points have sufficiently been addressed. I recommend publication of the article.

ok